# The Incidence, Aetiology and Clinical Course of Serious Infections Complicating Biological and Targeted Synthetic Disease-Modifying Antirheumatic Drug Therapy in Patients with Rheumatoid Arthritis in Tropical Australia

**DOI:** 10.3390/pathogens13110943

**Published:** 2024-10-29

**Authors:** Cody F. Price, John P. Wood, Ibrahim Ismail, Simon Smith, Josh Hanson

**Affiliations:** 1Department of Medicine, Cairns Hospital, Cairns, QLD 4870, Australia; cody.price@ths.tas.gov.au (C.F.P.); john.wood4@health.qld.gov.au (J.P.W.); ibrahim.ismail@health.qld.gov.au (I.I.); simon.smith2@health.qld.gov.au (S.S.); 2College of Medicine and Dentistry, James Cook University, Cairns, QLD 4878, Australia; 3The Kirby Institute, University of New South Wales, Kensington, NSW 2052, Australia

**Keywords:** infectious diseases, immunosuppression, tropical medicine, rheumatoid arthritis, Aboriginal and Torres Strait Islander peoples, tropical Australia

## Abstract

**Introduction**: Patients receiving biological and targeted synthetic disease-modifying antirheumatic drugs (b/tsDMARDs) for rheumatological conditions are at an increased risk of serious, potentially life-threatening, infection. However, the incidence, aetiology, and clinical course of serious infection in patients receiving b/tsDMARDs in tropical settings are incompletely defined. **Methods**: We retrospectively reviewed all patients with rheumatoid arthritis receiving b/tsDMARDs between October 2012 and October 2021, at Cairns Hospital in tropical Australia. The incidence, aetiology, and clinical course of serious infections (those requiring admission to hospital or parenteral antibiotics) were determined. **Results**: 310 patients had 1468 patient years of b/tsDMARD therapy during the study period; 74/310 (24%) had 147 serious infections translating to an overall risk of 10.0 episodes of serious infection per 100 patient years. The respiratory tract (50/147, 34%) and skin (37/147, 25%) were the most frequently affected sites. A pathogen was identified in 59/147 (40%) episodes and was most commonly *Staphylococcus aureus* (24/147, 16%). Only 2/147 (1%) were confirmed “tropical infections”: 1 case of *Burkholderia pseudomallei* and 1 case of mixed *B. pseudomallei* and community-acquired *Acinetobacter baumannii* infection. Overall, 13/147 (9%) episodes of serious infection required Intensive Care Unit admission (0.9 per 100-patient years of b/tsDMARD therapy) and 4/147 (3%) died from their infection (0.3 per 100-patient years of b/tsDMARD therapy). The burden of comorbidity and co-administration of prednisone were the strongest predictors of death or a requirement for ICU admission. **Conclusions**: The risk of serious infection in patients taking b/tsDMARDs in tropical Australia is higher than in temperate settings, but this is not explained by an increased incidence of traditional tropical pathogens.

## 1. Introduction

Since the advent of infliximab in 1998 there has been a rapid increase in the number of effective therapies for rheumatological disease. There are now multiple classes of biologic disease-modifying anti-rheumatic drugs (bDMARDs) and targeted synthetic disease-modifying anti-rheumatic drugs (tsDMARDs) which include tumour necrosis factor-alpha inhibitors (TNFi), interleukin 6 receptor inhibitors (IL-6Ri), co-stimulation inhibitors, and Janus kinase inhibitors (JAKi) [1].

While biological and targeted synthetic disease-modifying antirheumatic drugs (b/tsDMARDs) have revolutionised the care of patients with rheumatological disease, their prescription significantly increases the risk of serious, sometimes life-threatening, infection [2,3]. A meta-analysis—that included 106 randomized trials of patients with rheumatoid arthritis (RA) receiving bDMARDs—found that when compared with conventional synthetic disease-modifying anti-rheumatic drugs (csDMARDs), standard-dose bDMARDs (OR 1.31, 95% CI 1.09–1.58) and high-dose bDMARDs (OR 1.90, 95% CI 1.50–2.39) were associated with an increased rate of serious infection [4]. This higher risk of infection necessitates a thorough assessment of the patient before bDMARD commencement, with careful screening and—in selected cases—immunisation, pre-emptive therapy, and the initiation of primary antimicrobial prophylaxis [4].

Australian guidelines provide recommendations for screening and prophylaxis for patients with non-malignant conditions prior to receiving immunomodulatory therapy with TNFi or rituximab, although they do yet provide specific recommendations for patients receiving IL-6Ri, co-stimulation inhibitors or JAKi, as clinical experience with these therapies is still limited [5]. The current guidelines suggest that patients receiving TNFi therapy or rituximab should be screened and—where appropriate—treated for latent tuberculosis infection (LTBI), hepatitis B, and *Strongyloides stercoralis. Pneumocystis jirovecii* prophylaxis is recommended for patients who are receiving a TNFi or rituximab in combination with any other immunosuppressive drugs. Suppressive therapy for herpes simplex infection also requires consideration in the appropriate clinical context, while dietary modifications are recommended to prevent *Listeria monocytogenes* infection.

The guidelines also recommend that patients at risk of developing melioidosis (the tropical infectious disease caused by *Burkholderia pseudomallei*) may also require antimicrobial prophylaxis against this life-threatening pathogen [5]. This is important because b/tsDMARDs are increasingly prescribed in tropical Australia for rheumatological conditions [6]. Published guidelines exist for the prevention of opportunistic infections in immunosuppressed patients living in tropical Australia and they have been a valuable resource for clinicians working in tropical locations, particularly as they have a greater focus on Aboriginal and Torres Strait Islander Australians (hereafter respectfully referred to, collectively, as First Nations Australians) who represent a larger proportion of the population in tropical Australia [7]. These guidelines recommend that patients receiving immunosuppressive therapy in tropical Australia have serological screening for *B. pseudomallei* infection and that they receive indefinite prophylaxis if they have positive serology or if they have a history of melioidosis. It is also suggested that clinicians consider primary prophylaxis for patients with negative *B. pseudomallei* serology who live in endemic areas during the wet season. Long-term *S. stercoralis* prophylaxis with ivermectin is also recommended for patients who live in—or visit—remote First Nations communities. However, these recommendations, published in 2003, were based on data collected in the Top End of the Northern Territory of Australia before the widespread use of b/tsDMARDs. The contemporary relevance—and cost-effectiveness—of the guidelines in tropical Australia outside the Northern Territory have not been established.

Immunocompromised individuals in Far North Queensland (FNQ) in tropical Australia are at risk from a variety of infections. The region has a significant incidence of melioidosis, leptospirosis, and rickettsial disease [8,9,10]; a variety of parasitic, fungal, and mycobacterial infections are also locally endemic [11,12,13,14,15,16]. Meanwhile “typical” bacterial infections—such as *Staphylococcus aureus* and *Streptococcus pyogenes*—have a higher incidence in the region than in other parts of Australia [17,18]. FNQ shares a maritime border with Papua New Guinea which also increases the risk of imported pathogens [19,20]. FNQ contains three of the seven most socioeconomically disadvantaged regions in the country, a disadvantage that increases the local incidence of important infectious diseases [21,22,23]. Over a third of the population lives in rural and remote areas, increasing their risk of exposure to environmental pathogens [17,24]. Finally, approximately 17% identify as First Nations Australians, a population that carries a disproportionate burden of infectious diseases in the region [25,26,27,28].

This retrospective cohort study aimed to define the incidence, aetiology, associations, and clinical course of serious infections in patients with RA receiving b/tsDMARD therapies in this unique region of tropical Australia. It was hoped that these data might be used to define the risk of serious infection in the immunosuppressed population in the region to inform optimal contemporary patient care.

## 2. Methods

This retrospective study was performed at Cairns Hospital, a tertiary referral 531-bed hospital in tropical Australia that serves a population of approximately 290,000 people, living across an area of about 380,000 km^2^. The Cairns Hospital Rheumatology Department is the sole prescriber of b/tsDMARDs for rheumatological diagnoses in the region and has a database of every patient prescribed b/tsDMARDs for a rheumatological condition; eligible patients were identified using this database.

Patients were eligible for the study if they had received b/tsDMARDs for RA between October 2012 and October 2021. The ten-year study period was chosen as it coincided with the expansion of the local rheumatology service and an increase in the prescription of b/tsDMARDs, but prior to the impact of significant COVID-19 transmission in the region [29]. Patients were excluded if they were children (age < 18 years), had another indication for b/tsDMARD therapy, or if the duration of b/tsDMARD therapy was <1 month.

Demographic data (age, gender, identification as a First Nations Australian, and residential address) were collected from medical records. Comorbidity data were recorded and quantified using the Charlson comorbidity index, with severe comorbidity defined as a score ≥ 5 [30]. Individual comorbidities were defined using the definitions of the Charlson comorbidity index. Patients living within 20 km of the Cairns central business district were said to have an urban residence, while those living >100 km away were said to live remotely. Patients were defined as seropositive if they had a serum rheumatoid factor ≥ 20 IU/mL or serum antibodies to cyclic citrullinated peptide ≥ 6 IU/mL.

Under Australia’s Pharmaceutical Benefits Scheme, patients are only eligible for subsidised b/tsDMARD therapy if they have trialled at least 2 csDMARDs over a 6-month period and have persistent disease activity as determined clinically by their treating rheumatologist. Persistent disease activity is defined by the presence of at least 20 tender or swollen small joints or 4 or more large joints on examination together with raised inflammatory markers (CRP > 15 mg/L and or ESR > 25 mm/hour). The patients’ joint count prior to commencement of b/tsDMARD therapy (the number of tender and swollen joints) was recorded as a proxy for their underlying disease severity.

The nature and duration of the patients’ prior therapy for their rheumatological disease were also documented, specifically treatment with glucocorticoids, csDMARDs, and b/tsDMARDs including TNFi (adalimumab, golimumab, etanercept, certolizumab, and infliximab), JAKi (baricitinib and tofacitinib), the CLTA4 inhibitor abatacept, the IL-6 inhibitor tocilizumab, and the B-cell depletion monoclonal antibody rituximab.

Medical records, laboratory, and radiology data were examined for details of pre-immunosuppression screening for tuberculosis (a Mantoux test, TB-specific interferon gamma release assay (IGRA), and chest X-ray), hepatitis B (hepatitis B surface antigen (HBsAg) and hepatitis B core antibody (HBcAb)), melioidosis (*B. pseudomallei* serology), and strongyloidiasis (*S. stercoralis* serology). Medical records were also reviewed for a prior diagnosis of tuberculosis, hepatitis B, *Pneumocystis jirovecii* pneumonia, melioidosis, or strongyloidiasis or documented prescription of prophylaxis for—or pre-emptive treatment of—any of these infections.

The incidence, aetiology, and clinical course of serious infections were determined for the duration of time that patients were receiving locally prescribed b/tsDMARD therapy between October 2012 and October 2021. Serious infection was defined as one requiring admission to hospital, one requiring parenteral antimicrobials in the hospital’s outpatient parenteral antimicrobial program, or one requiring parenteral antimicrobials during a hospital admission for another indication. Individual admissions were identified in the patients’ electronic medical records. An instance of a single infection requiring multiple admissions was counted as a single episode of infection. The patients’ medical records were reviewed to determine the site of the infection; if an infection involved multiple organs—or did not localise to a single organ—it was defined as a systemic infection.

Individual pathogens were identified in AUSLAB, Queensland’s statewide electronic laboratory database. A tropical pathogen was defined as an infectious agent that is endemic to—and is isolated principally in—tropical and subtropical regions [31]. Each instance of serious infection was reviewed, and relevant data were collected, including the duration of b/tsDMARD therapy during the study period prior to admission, other immunosuppressive therapy at the time of admission, the clinical diagnosis for their infection, and the microbiological diagnosis, where one was confirmed. The presence of comorbidities or lifestyle factors that may have contributed to the development of infection, the duration of hospitalization, intensive care unit (ICU) admission, and death before hospital discharge were also recorded.

### 2.1. Statistical Analysis

Data were entered into an electronic database (Microsoft Excel 2016, Microsoft, Redmond, WA, USA), de-identified, and analysed using statistical software (Stata version 18.0, StataCorp LLC, College Station, TX, USA). As two of the three continuous variables that we examined had a non-parametric distribution (Charlson comorbidity index and joint count) the median and interquartile range (IQR) were presented for all continuous variables. Groups were analysed using logistic regression, the chi-squared, Fisher’s exact, or the Wilcoxon rank sum test, where appropriate. The risk of infection was determined using a Cox proportional hazards model. Multivariate analysis of selected groups was performed using a backward stepwise approach; variables with a *p* < 0.10 in univariate analysis were selected for the model, although only those with a *p*-value of <0.05 were retained. Trends over time were determined using logistic regression. If individuals were missing data, they were not included in analyses that evaluated those variables.

### 2.2. Ethical Approval

The Far North Queensland Human Research Ethics Committee provided ethical approval for the study (EX/2023/QCH/95366—1706 QA). As the data were retrospective and de-identified, the Committee waived the requirement for informed consent.

## 3. Results

There was a total of 310 patients who were eligible for the study (Appendix A), 204/310 (66%) commenced b/tsDMARD therapy in Cairns during the study period, 84/310 initiated b/tsDMARD therapy in Cairns prior to the study period, and 22/310 (7%) had commenced b/tsDMARD therapy at another centre previously. Patients were treated with csDMARDs for a median (IQR) of 4 (2–9) years prior to commencement of b/tsDMARD therapy. The patients’ median (interquartile range (IQR)) age at their first prescription of b/tsDMARD therapy in Cairns during the study period was 56 (47–63) years. The patients were predominantly female (225/310 (73%)), 180/310 (58%) had an urban residence while 59/310 (19%) lived remotely. Of 292 (94%) with available serological data, 226 (77%) had seropositive disease (Table 1). The median (IQR) Charlson comorbidity index of the patients in the cohort was 2 (1–3); 31/310 (10%) had severe comorbidity.

### 3.1. First Nations Australian Patients

There were 33/310 (11%) who identified as a First Nations Australian. First Nations Australian patients were younger than non-First Nations Australian patients at the time of their first b/tsDMARD prescription in Cairns during the study period (median (IQR) 52 (41–57) years versus 57 (47–63) years, *p* = 0.01) and were more likely to live remotely (12/33 (36%) versus 47/277 (17%), *p* = 0.007). They also had greater comorbidity than non-First Nations Australian patients (median (IQR) Charlson comorbidity index: of 2 (1–4) versus 1 (1–3), *p* = 0.04). A greater proportion of First Nations Australians were cigarette smokers (16/25 (64%) versus 42/207 (20%), *p* < 0.0001) among the 232/310 (68%) individuals who had this data available. A greater proportion of First Nations Australians had seropositive RA, but this difference did not reach statistical significance (28/31 (90%) versus 198/261 (76%), *p* = 0.07). There was also no difference in the median (IQR) baseline joint count before b/tsDMARD prescription (24 (7–31) versus 24 (12–38), *p* = 0.91)

### 3.2. b/tsDMARD Therapy

Patients in the cohort were treated with b/tsDMARDs in Cairns for a median (IQR) of 3.8 (2.0–7.6) years during the study period, representing 1468 patient years of exposure. The initial b/tsDMARD was most commonly a TNFi: adalimumab in 130/310 (42%), etanercept in 72/310 (23%), golimumab in 31/310 (10%), certolizumab in 22/310 (7%), and infliximab in 12/310 (4%). The initial non-TNFi b/tsDMARD therapy was tocilizumab in 15/310 (5%), abatacept in 13/310 (4%), rituximab in 10/310 (3%), and tofacitinib in 1/310 (0.3%).

However, many patients changed their b/tsDMARD during the study period. At the conclusion of the study period, adalimumab was the treatment for 57/310 (18%), tocilizumab for 46/310 (15%), etanercept for 42/310 (14%), rituximab for 41/310 (13%), golimumab for 18/310 (6%), certolizumab for 14/310 (5%), abatacept for 10/310 (3%), tofacitinib for 9/310 (3%), baricitinib for 2/310 (0.7%), and infliximab for 2/310 (0.6%). b/tsDMARDs were ceased in 37/310 (12%) and there were 32/310 (10%) who had moved to a different location/prescriber or who were lost to follow-up during the study period.

Patients prescribed rituximab during the study period had been taking biologic therapy for longer than those who had not received this agent (median (IQR) 5.5 (3.0–10.0) versus 3.0 (1.8–7.3) years), *p* = 0.0005) and they had greater comorbidity (median (IQR) Charlson comorbidity index 3 (1–4) versus 1 (1–2), *p* = 0.0007).

### 3.3. Screening for Infection Prior to Initiation of b/tsDMARD Therapy

A total of 277/310 (89%) had an accessible CXR result, including 143/146 (98%) starting b/tsDMARDs in Cairns in the last 5 years of the study. There were 259/310 (84%) who had an accessible Mantoux test and/or a TB-specific IGRA result, including 145/146 (99%) starting b/tsDMARDs in Cairns in the last 5 years of the study (Appendix A).

The CXR was abnormal in 21/277 (8%). LTBI treatment was initiated on 10/21 with abnormal chest X-rays, 4/10 (40%) of whom had a positive IGRA or Mantoux. Of the 256 patients with normal chest X-rays, 19 (7%) had an abnormal immunological test of whom 8 (42%) received LTBI treatment (Appendix A).

No patient had a history of chronic hepatitis B prior to commencing b/tsDMARDs therapy. An HBsAg result was available in 285/310 (92%), including all 146 individuals starting therapy in the last 5 years of the study. An HBcAb test result was accessible in 117/310 (38%), including 70/146 (48%) starting therapy in the last 5 years of the study; 10/117 (9%) HBcAb tests were positive. No patients received prophylaxis against hepatitis B reactivation. Only one HBcAb-positive patient was prescribed rituximab; this patient was also hepatitis B surface antibody positive and underwent regular monitoring during the study period without initiating antiviral therapy. The number of patients who underwent HBcAb testing increased during the study period (odds ratio (OR) (95% confidence interval (95% CI): 1.14 (1.04–1.26), *p* = 0.007 (Appendix A).

A *S. stercoralis* serology result was accessible in 81/310 (26%), 5/81 (6%) were positive, 4 (80%) of whom received ivermectin prior to b/tsDMARD commencement, but no patients had ongoing prophylaxis. The number of patients with an accessible *S. stercoralis* serology result increased during the study period (odds ratio (OR) (95% confidence interval (95% CI): 1.39 (1.25–1.55), *p* < 0.001 (Appendix A).

No patient had a history of prior melioidosis. *B. pseudomallei* serology results were available in 10/310 (3%) and were negative in all. Australian guidelines would have suggested that 281/310 (91%) satisfied the criteria for *Pneumocystis jirovecii* prophylaxis at some point in the study period [5], but only 1 patient was prescribed prophylaxis (in the context of high-dose corticosteroid therapy).

### 3.4. Incidence and Timing of Infection

There were 74/310 (24%) individuals who had 147 episodes of serious infection during the study period, translating to an overall risk of infection of 10.0 per 100 patient years of b/tsDMARD therapy. These 74 individuals had a median (IQR) number of 1 (1–3) serious infections. There were 13/310 (4%) individuals who had 3 or more episodes of serious infection, and these patients accounted for 75/147 (51%) of the serious infection episodes.

The median (IQR) duration of b/tsDMARD therapy in Cairns during the study period at the time of the 74 patients’ first infection in the study period was 2.2 (1.2–4.8) years. Of the 204 patients initiating b/tsDMARD therapy in Cairns during the study period, 41 (20%) developed a serious infection after a median (IQR) duration of b/tsDMARD therapy of 1.5 (0.8–4.0) years (Figure 1). Patients starting b/tsDMARD therapy in Cairns during the study period were no more likely to develop a serious infection than individuals already taking b/tsDMARD therapy at the start of the study period (41/204 (20%) versus 33/106 (31%), (hazard ratio (HR) (95% confidence interval (CI)): 1.27 (0.79–2.05), *p* = 0.33).

Univariate analysis of the association between demographic and clinical characteristics of the patients who developed any infection and multiple infections is presented in Table 1 and Appendix A, respectively. None of the different classes of b/tsDMARD therapy had a statistically significant association with the development of infection (Appendix A).

In multivariate analysis, three factors were independently associated with the development of infection during the study period: age at study entry (HR (95% CI): 1.03 (1.01–1.06), *p* = 0.007), identification as a First Nations Australian (HR (95% CI): 3.05 (1.50–6.19), *p* = 0.002), and the baseline joint count (HR (95% CI): 1.04 (1.01–1.07), *p* = 0.008) (Figure 2). The lower incidence of multiple infections precluded reliable multivariate analysis of the factors that were independently associated with multiple infections.

### 3.5. Site and Aetiology of Infection

The most common sites of infection were the respiratory tract (50/147 (34%)) and the skin (37/147 (25%)). The patient had a systemic infection in 19/147 (13%), the gastrointestinal tract was involved in 20/147 (14%) and bones or joints were involved in 12/147 (8%). The urinary tract was involved in 6/147 (4%) and the eye was involved in 2/147 (1%).

Pathogens were identified in 59/147 (40%) episodes; in 51/59 (86%), there was a single pathogen, and in 8/59 (14%), more than 1 pathogen was identified. Bacterial pathogens were identified in 56/59 (95%) episodes and were most commonly *S. aureus* (24/56 (43%), methicillin-sensitive *S. aureus* in 20/24 (83%), and methicillin-resistant *S. aureus* in 4/24 (17%)). There was no association between the isolation of *S. aureus* and age (*p* = 0.62), First Nations Australian status (*p* = 1.0), corticosteroid use (*p* = 0.12), or comorbidity (*p* = 0.53).

Tropical pathogens were identified in only 2/147 (1%) episodes; there was 1 case of disseminated melioidosis (in a non-First Nations Australian patient on rituximab who also had a myelodysplastic syndrome) and 1 case of community-acquired pneumonia and osteomyelitis where both *Acinetobacter baumannii* and *B. pseudomallei* were isolated in a First Nations Australian man with functional hyposplenism and chronic lung disease who was taking etanercept, methotrexate, and hydroxychloroquine. His case has been reported previously [32].

Other bacterial pathogens isolated in monomicrobial infections included *Escherichia coli* in 5 cases (1 of which was an extended-spectrum beta-lactamase producer), *Streptococcus pneumoniae* in 3 cases, *Haemophilus influenzae* in 2 cases, *Campylobacter* sp. in 2 cases, *Clostridioides difficile* in 1 case, *Coxiella burnetii* in 1 case, *Enterobacter cloacae* in 1 case, *Morganella morganii* in 1 case, *Mycobacterium intracellulare* in 1 case, *Peptostreptococcus* sp. in 1 case, *Pseudomonas aeruginosa* in 1 case, and *Staphylococcus capitis* in 1 case. There was 1 case in which a fungal pathogen was identified—*Curvularia* sp.—as the causative agent of recurrent dacryocystitis. Despite the tropical setting of the study, this was the only episode where a fungal pathogen or parasite was identified.

In 6/59 (10%) episodes with a microbiological diagnosis, a virus was identified (influenza A in 3 cases, parainfluenza in 1 case, rhinovirus in 1 case, and herpes zoster in one case). There were no confirmed or suspected cases of *P. jirovecii* pneumonia, strongyloidiasis, cryptococcosis, leptospirosis, rickettsial disease, or hepatitis B reactivation during the study period. There was 1 case of suspected disseminated tuberculosis, however, there was no positive culture of *Mycobacterium tuberculosis*, the diagnosis was based on clinical and histological findings. This patient—who had migrated to Australia from the Philippines—had a normal chest X-ray and negative Mantoux test before b/tsDMARD initiation.

### 3.6. Clinical Course

The median (IQR) duration of hospitalisation for the 147 serious infections was 5 (2–9) days. ICU admission was required in 13/147 (9%) episodes (in 11 individual patients) translating to an overall risk of infection requiring ICU admission of 0.9 per 100 patient years of b/tsDMARD therapy; the median (IQR) duration of ICU admission was 3 (2–6) days. There were 24/310 (8%) people who died during the study period; infection was deemed the cause of death in 4/24 (17%) equating to a risk of 0.3 per 100 patient years. The characteristics of the patients whose serious infection resulted in death and/or ICU admission are presented in Table 2. The burden of comorbidity and the co-prescription of glucocorticoids were associated with death or ICU admission (Table 3). Both patients with a tropical infection survived their infection.

## 4. Discussion

Patients receiving b/tsDMARD therapy for RA in this region of tropical Australia have an incidence of serious infection that is greater than three times the rate seen in temperate settings [2,3,33]. However, this higher incidence of serious infections is not explained by a greater burden of tropical pathogens. Although serious infection was a common complication of local b/tsDMARD therapy, the rate of infection-related death was no greater than that seen in temperate settings [34]. This is likely a function of universal access to sophisticated health care in Australia’s well-resourced public health system [35].

Although First Nations Australians represented only 10% of the cohort, they experienced over a third of the serious infections during the study period. First Nations Australians in the FNQ region suffer from a disproportionate burden of infectious diseases, a disparity which is largely explained by the social determinants of health and the disadvantage that many First Nations Australians in the region continue to experience [22,23,27,28,36]. Although First Nations Australians in the study’s cohort were younger than the non-First Nations Australians, they had greater comorbidity increasing their risk of serious infection [37]. First Nations Australians in the study’s cohort were also more than twice as likely to live in remote locations where there is less access to high-quality housing and health and other services that might also be anticipated to reduce the incidence of preventable infections [38,39].

However, despite the greater incidence of serious infection in the study, its unique tropical setting, the remote residence of many of the patients and a significant proportion of First Nations Australians, it was striking how similar the study’s other findings were to studies performed in temperate, metropolitan settings in Europe and North America. The most common sites of infection—the respiratory tract and the skin and soft tissues—were similar to the sites affected in patients with RA in temperate settings not receiving b/tsDMARDs [40,41]. Bone and joint infections were relatively common as has been noted in other cohorts of patients taking bDMARDs [42]. An association between age and comorbidity—particularly chronic lung disease—echoes that seen in other cohorts [3,33,43], as does the association between corticosteroid therapy and life-threatening infection [44,45,46]. Serious infection was also independently associated with baseline joint count, a crude proxy of the underlying severity of the rheumatological disease and, potentially, the immunosuppressive therapy that the patient received subsequently [47].

A similarly high rate of serious infection of 11.7 per 100 patient years was reported in a cohort of patients taking b/tsDMARD therapy—for a variety of rheumatological conditions—in the neighbouring tropical region of Townsville, with the respiratory tract and the skin and soft tissue also the commonest site of infections [48]. Infection in the Townsville patients was, as in our cohort, associated with increasing age, chronic pulmonary disease, and the use of glucocorticoids, although the report did not detail the aetiology and clinical course of the infections. The Townsville report also identified that rituximab was also associated with the development of infection. Rituximab is a second line agent for rheumatological disease and may, therefore, be a marker of more difficult to control disease and greater immunosuppression [49,50]. In our and other cohorts, patients receiving rituximab have greater comorbidity and have been taking b/tsDMARD therapy for longer [51]. Rituximab’s mechanism of effect—B cell depletion—is also different to the other b/tsDMARDs in the study and it has a longer duration of action [52]. Although we did not see an independent association between rituximab therapy and serious infection in our cohort, the relatively common use of rituximab as second-line bDMARD therapy in our cohort (Appendix A) may have contributed to the observation that there was no decrease in the incidence of serious infections in the study period over time, a finding which has been described in previous series [33].

Reactivation of LTBI is one of the most characteristic complications of bDMARD therapy and the FNQ region has a tuberculosis incidence of 14.8 per 100,000 population, almost three times the national Australian rate [14]. Indeed, over 6% of the patients in this cohort were treated for LTBI prior to commencement of bDMARDs and there was one case of probable disseminated tuberculosis—with negative baseline screening—which complicated therapy. However, while non-tuberculous mycobacterial infections are seen more commonly in patients receiving bDMARD therapy, there was only one case of non-tuberculous mycobacterial infection in the cohort despite the significant local burden of these infections [15,53].

The absence of tropical pathogens in the series was notable and echoes other studies that have examined the microbiological aetiology of common clinical presentations in the region [24,27,28,54,55,56]. Despite the study’s tropical location, other tropical infections associated with bDMARD therapy including cryptococcosis, nocardiosis, salmonellosis, and actinomycosis were not isolated, although all these pathogens are locally endemic [13,57,58,59]. Additionally, there were no cases of listeriosis or legionellosis during the study [60,61]. Despite a significant local prevalence of chronic hepatitis B infection [25], there were no cases of hepatitis B reactivation. Most patients underwent HBsAg testing, and all were negative. The number of patients who had HBcAb testing performed was much lower although testing did increase during the study period, which may reflect increasing awareness amongst local clinicians that HBsAg negative HBcAb positive patients are at risk of hepatitis B reactivation during B cell depleting therapy with rituximab as well as with TNFi therapy [62].

National guidelines recommend trimethoprim-sulfamethoxazole (TMP-SMX) prophylaxis for *P. jirovecii* pneumonia in patients taking TNFi and rituximab, if they are treated with other immunosuppressive agents or have T-cell defects or significant lymphopenia [5]. However, despite only 1 patient in this cohort receiving TMP-SMX prophylaxis, there were no cases of *P. jirovecii* pneumonia. This is consistent with international observations that TMP-SMX prophylaxis is not a high-value intervention in patients receiving bDMARD therapy in the absence of significant concomitant immunosuppression, in particular, the use of high-dose glucocorticoids [63]. TMP-SMX may also reduce the incidence of bacterial infections including *S. aureus*—the most commonly isolated pathogen in our cohort—and *B. pseudomallei* [64,65,66], although this needs to be balanced against the risk of significant adverse reactions, which are more common in patients who are also prescribed methotrexate [67,68,69,70].

Although clinicians in the Northern Territory of Australia routinely test *B. pseudomallei* serology, prior to commencing significant immunosuppressive therapies and prescribe prophylactic TMP/SMX for those who test positive [7], this strategy has not been employed in FNQ. Although there has been a significant recent increase in the incidence of melioidosis in FNQ—and immunosuppression contributes to the development of the infection in almost 20% of local cases [8,71]—there were only 2 episodes of non-fatal melioidosis in our cohort and both patients had other traditional risk factors for the disease in addition to their bDMARD therapy.

Although few patients were screened for *S. stercoralis,* the positivity rate of 6% in those tested was lower than the rates of other seropositivity studies performed in the FNQ region [11]. Despite infrequent pre-emptive treatment and no ongoing prophylaxis, there were no cases of symptomatic strongyloidiasis or strongyloidiasis hyperinfection in our cohort. The low screening rates may reflect local clinicians’ perception that strongyloidiasis is now uncommon in FNQ [11]. Additionally, while glucocorticoids are a well-established risk factor for strongyloidiasis hyperinfection due to their effect on host anti-helminth immunity, the risk in people taking b/tsDMARD monotherapy is less well defined [72]. However, given the life-threatening potential of the infection in immunocompromised patients and the safety and ease of ivermectin therapy [73], there is no reason to vary from the recommendation to universally screen, treat positive serology, and provide ongoing prophylaxis as recommended in current guidelines until more longitudinal data about *S. stercoralis* prevalence becomes available [5].

Our study has many limitations. Its retrospective nature precluded comprehensive data collection in all patients. Screening laboratory tests were often performed many years before the study, at other sites and by private laboratories and the results of this testing were not always accessible. We present only a crude analysis of the incidence of serious infection. It was not possible to adjust completely for host factors (different comorbidities, behaviours, and access to care), disease factors (seropositivity, duration of disease, prior csDMARD therapies), and to control for the degree of immunosuppression (number of concurrent agents including csDMARDs and glucocorticoids and their doses). There are significant differences in the mechanism of action—and risk of infection—with different b/tsDMARDs [42,74,75,76]; our study was unable to completely control for this. The frequent adjustment of b/tsDMARD therapy during the study also precluded detailed analysis of the infection risk of individual b/tsDMARD agents. By examining only infections that necessitated hospitalisation or intravenous antibiotic therapy, we will have underestimated the impact of other infections such as herpes zoster which can be associated with significant short- and long-term morbidity [77]. We were also not able to access the vaccination histories of all the patients, which precluded analysis of any impact that immunisation may have had on the incidence of vaccine-preventable diseases such as influenza and pneumococcal disease. Our findings are not necessarily generalizable to other tropical settings—in Australia or internationally—where the risk of different opportunistic infections is likely to vary which will, accordingly, impact on optimal strategies to prevent them [65,78,79,80,81].

However, the study provides contemporary data for clinicians managing rheumatological conditions with b/tsDMARD therapy in tropical settings, highlighting the common pathogens and the factors that have the greatest association with serious infection. As strategies to manage rheumatological disease evolve, future prospective studies might examine how different management algorithms can further refine approaches to reducing the incidence of serious life-threatening infection while also controlling the underlying rheumatological process. Tailored, individualized approaches that take into consideration age, comorbidity, the degree of immunosuppression, and local variations in the risk of exposure to different potential pathogens are likely to be necessary.

## 5. Conclusions

In this region of tropical Australia, the incidence of serious infection complicating b/tsDMARD therapy for RA was over three times greater than that reported from temperate settings and disproportionately affected First Nations Australians. However, despite the tropical setting of the study, serious infections most commonly involved the respiratory tract and skin and soft tissues and were due to common bacterial pathogens.

## Figures and Tables

**Figure 1 pathogens-13-00943-f001:**
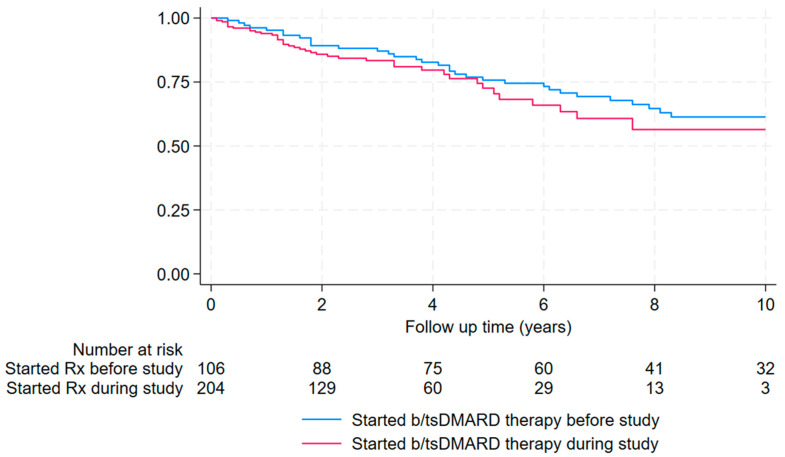
Time to first serious infection between October 2012 and October 2021, stratified by whether b/tsDMARD therapy was initiated during the study period or whether the patient was already receiving b/tsDMARD therapy at the start of the study.

**Figure 2 pathogens-13-00943-f002:**
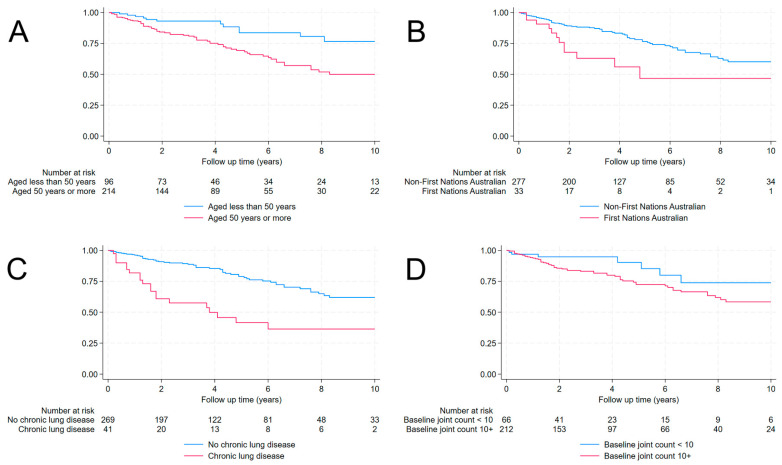
Time to first serious infection between October 2012 and October 2021, stratified by the factors that were most associated with the development of infection. (**A**): Age; (**B**): First Nations Australian status; (**C**): Underling lung disease; (**D**): Joint count at the initiation of b/tsDMARD therapy.

**Table 1 pathogens-13-00943-t001:** The association between the baseline demographic and clinical characteristics of the patients and their subsequent development of serious infection while receiving b/tsDMARD therapy.

	All *n* = 310	No Serious Infection *n* = 236	Any Serious Infection *n* = 74	Hazard Ratio (95% CI)	*p*
**Age (years) ^a^**	**56 (47–63)**	**54 (45–63)**	**58 (53–65)**	**1.04 (1.01–1.06)**	**0.001 ^b^**
Female Gender	225 (73%)	169 (72%)	56 (76%)	1.20 (0.70–2.04)	0.51
**First Nations Australian**	**33 (11%)**	**21 (9%)**	**12 (16%)**	**2.43 (1.30–4.54)**	**0.005 ^b^**
Urban address	180 (56%)	141 (60%)	39 (53%)	0.79 (0.50–1.25)	0.32
Remote address	59 (19%)	41 (17%)	18 (24%)	1.49 (0.88–2.54)	0.12
**Charlson Comorbidity Index**	**2 (1–3)**	**1 (1–2)**	**2 (1–4)**	**1.14 (1.04–1.26)**	**0.005 ^b^**
Severe Comorbidity (≥5)	31 (10%)	18 (8%)	13 (18%)	1.73 (0.95–3.15)	0.07
**Cardiac disease**	**29 (9%)**	**17 (7%)**	**12 (16%)**	**1.86 (1.00–3.46)**	**0.049 ^b^**
**Lung disease**	**41 (13%)**	**21 (9%)**	**20 (27%)**	**3.28 (1.96–5.48)**	**<0.001 ^b^**
Diabetes mellitus	19 (6%)	13 (6%)	6 (8%)	1.80 (0.78–4.15)	0.17
Neurological disease	11 (4%)	8 (3%)	3 (4%)	1.02 (0.32–3.23)	0.98
Renal disease	3 (1%)	2 (1%)	1 (1%)	1.02 (0.14–7.34)	0.99
Liver disease	4 (1%)	3 (1%)	1 (1%)	1.12 (0.16–8.07)	0.91
Seropositive disease ^c^	226/292 (77%)	175/224 (78%)	51/68 (75%)	0.93 (0.53–1.61)	0.79
**Joint count ^c,d^**	**24 (10–28)**	**23 (8–27)**	**26 (22–31)**	**1.04 (1.01–1.07)**	**0.005 ^b^**
Glucocorticoids at baseline ^c^	119/284 (42%)	99/220 (45%)	20/64 (31%)	0.85 (0.50–1.44)	0.54
Started b/tsDMARD during the study	204 (66%)	163 (69%)	41 (55%)	1.27 (0.79–2.05)	0.33
TNFi ever	281 (91%)	214 (91%)	67 (91%)	1.20 (0.55–2.63)	0.64
Adalimumab ever	163 (53%)	121 (51%)	42 (57%)	0.94 (0.59–1.49)	0.78
Golimumab ever	45 (15%)	32 (14%)	13 (18%)	1.49 (0.82–2.72)	0.19
Etanercept ever	126 (41%)	99 (42%)	27 (36%)	0.71 (0.44–1.14)	0.15
Certolizumab ever	35 (11%)	27 (11%)	8 (11%)	1.09 (0.52–2.28)	0.81
Infliximab ever	17 (5%)	12 (5%)	5 (7%)	0.93 (0.37–2.30)	0.87
JAKi ever	17 (5%)	14 (6%)	3 (4%)	0.80 (0.25–2.53)	0.70
Baricitinib ever	2 (1%)	2 (1%)	0	-	-
Tofacitinib ever	16 (5%)	13 (6%)	3 (4%)	0.84 (0.26–2.65)	0.76
Abatacept ever	35 (11%)	25 (11%)	10 (14%)	0.86 (0.44–1.68)	0.67
Tocilizumab ever	98 (32%)	68 (29%)	30 (41%)	1.10 (0.69–1.75)	0.69
Rituximab ever	57 (18%)	35 (15%)	22 (30%)	1.50 (0.91–2.47)	0.12

Numbers represent the median (interquartile range) or absolute number (%). TNFi: Tumour necrosis factor inhibitor; JAKi: Janus kinase inhibitor. ^a^ At the time b/tsDMARD therapy was first prescribed in Cairns during the study period. ^b^ Included in multivariate analysis. ^c^ Incomplete data: Due to the retrospective nature of the study and the commencement of b/tsDMARD therapy at other sites, it was only possible to determine seropositivity in 292/310 (94%), a baseline joint count in 278/310 (90%) and the prescription of steroids at baseline in 284/310 (92%). ^d^ Joint count includes both tender and swollen joints.

**Table 2 pathogens-13-00943-t002:** Demographic, clinical, and microbiological characteristics of the 4 patients who died and the additional 12 patients who were admitted to the intensive care unit and survived during the study.

Age, Sex	First Nations Australian	Remote Residence	Charlson Comorbidity Index	Clinical Syndrome	Microbiological Isolate	Other Immunosuppression at Time of Admission	ICU Admission	B/tsDMARD at Time of Admission	Outcome
46 F	No	No	3	Cellulitis leading to bacteraemia and systemic infection	MRSA	HCQ and prednisone	Yes	Adalimumab	Died
72 F	No	No	6	Cellulitis	None	LFL, HCQ, and prednisone	No	Etanercept	Died
66 F	No	No	4	IE COPD	None	Prednisone	No	Adalimumab	Died
67 M	No	No	4	Neutropenic sepsis	None	Prednisone	No	Rituximab	Died
61 M	Yes	No	3	Community-acquired pneumonia and osteomyelitis	*Burkholderia pseudomallei* and *Acinetobacter baumannii*	MTX, HCQ	Yes	Etanercept	Survived
61 F	Yes	No	6	Septic shock with unclear source	None	LFL and prednisone	Yes	Infliximab	Survived
61 F	Yes	No	6	Septic shock with unclear source	None	LFL and prednisone	Yes	Certolizumab	Survived
61 F	No	No	4	Community-acquired pneumonia		MTX and prednisone	Yes	Tocilizumab	Survived
57 F	Yes	Yes	8	IE COPD	*Streptococcus pneumoniae*	MTX, HCQ, and prednisone	Yes	Etanercept	Survived
57 F	Yes	Yes	8	Urinary tract infection	*Escherichia coli*	MTX	Yes	Etanercept	Survived
60 F	No	Yes	7	Hospital-acquired pneumonia	Influenza A	None	Yes	Tocilizumab	Survived
62 F	No	No	4	Colitis	*Clostridioides difficile*	HCQ	Yes	Tocilizumab	Survived
71 F	No	No	5	Diverticulitis	None	HCQ and prednisone	Yes	Tocilizumab	Survived
64 F	No	No	4	Community-acquired pneumonia	*Streptococcus pneumoniae*	prednisone	Yes	Adalimumab	Survived
47 F	No	No	4	Prosthetic joint infection	MSSA	MTX and prednisone	Yes	Rituximab	Survived
69 F	No	No	4	Cholecystitis	Polymicrobial	MTX and HCQ	Yes	Tocilizumab	Survived

ICU: Intensive Care Unit; F: female; M: male; IE COPD: infective exacerbation of chronic pulmonary disease; CAP: community-acquired pneumonia; HAP: hospital-acquired pneumonia; MRSA: methicillin-resistant *Staphylococcus aureus*; MSSA: methicillin-sensitive *Staphylococcus aureus*; HCQ: hydroxychloroquine; LFL: leflunomide; MTX: methotrexate.

**Table 3 pathogens-13-00943-t003:** In cases of serious infection, risk factors at the time of presentation for the combined endpoint of Intensive Care Unit Admission and death.

	All *n* = 147	No ICU/Death *n* = 131	ICU/Death *n* = 16	*p*
Age at admission (years)	61 (56–69)	61 (56–69)	61 (58–67)	0.77
Female gender	112 (76%)	98 (75%)	14 (88%)	0.36
First Nations Australian	50 (34%)	45 (34%)	5 (31%)	1.0
Urban address	57 (39%)	49 (37%)	8 (50%)	0.42
Remote address	53 (36%)	50 (38%)	3 (19%)	0.17
Seropositive rheumatoid arthritis	106/122 (87%)	94/110 (85%)	12/12 (100%)	0.36
**Charlson Comorbidity Index**	**3 (2–5)**	**3 (2–4)**	**4 (4–6)**	**<0.001**
Severe comorbidity (≥5)	38 (26%)	31 (24%)	7 (44%)	0.13
Other contributing factors ^a^	81 (55%)	71 (54%)	10 (63%)	0.60
On adalimumab at admission	39 (27%)	36 (27%)	3 (19%)	0.56
Golimumab at admission	8 (5%)	8 (6%)	0	0.60
Etanercept at admission	27 (18%)	23 (18%)	4 (25%)	0.50
Certolizumab at admission	7 (5%)	6 (5%)	1 (6%)	0.56
Infliximab at admission	4 (3%)	3 (2%)	1 (6%)	0.37
Tofacitinib at admission	3 (2%)	3 (2%)	0	1.0
Abatacept at admission	2 (1%)	2 (2%)	0	1.0
Tocilizumab at admission	38 (26%)	33 (25%)	5 (31%)	0.56
Rituximab previously	24 (16%)	20 (15%)	4 (25%)	0.30
Methotrexate at admission	77 (52%)	71 (54%)	6 (38%)	0.29
Leflunomide at admission	24 (16%)	20 (15%)	4 (25%)	0.30
Sulfasalazine at admission	3 (2%)	3 (2%)	0	1.0
Hydroxychloroquine at admission	51 (35%)	44 (34%)	7 (44%)	0.42
**Glucocorticoids at admission**	**44 (30%)**	**33 (25%)**	**11 (69%)**	**0.001**
**Prednisone dose (milligrams) at admission**	**0 (0–4)**	**0 (0–1)**	**4 (0–5)**	**0.001**

Numbers represent the median (interquartile range) or absolute number (%). ICU: Intensive Care Unit. ^a^ Other contributing factors included comorbidity that increased the risk of this episode infection including chronic obstructive pulmonary disease, bronchiectasis, interstitial lung disease, chronic skin ulcer, trauma, surgery, urinary catheterisation, diabetes, hyposplenism, neutropenia, and malignancy.

## Data Availability

Data cannot be shared publicly because of the Queensland Public Health Act 2005. Data are available from the Far North Queensland Human Research Ethics Committee (contact via email FNQ_HREC@health.qld.gov.au) for researchers who meet the criteria for access to confidential data.

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
