# Peer review of "The Incidence, Aetiology and Clinical Course of Serious Infections Complicating Biological and Targeted Synthetic Disease-Modifying Antirheumatic Drug Therapy in Patients with Rheumatoid Arthritis in Tropical Australia"

_pathogens, 2024, doi:10.3390/pathogens13110943_

Round 1

Reviewer 1 Report

Comments and Suggestions for Authors

Dear Authors,

I have read your manuscript with great interest. In my opinion it is well prepared.

My only question regards vaccination status among the participants, especially influenza and pneumococcal infection. If data are not known put it into limitation of the study.

Author Response

Reviewer's comments

1. Dear Authors,

I have read your manuscript with great interest. In my opinion it is well prepared.

My only question regards vaccination status among the participants, especially influenza and pneumococcal infection. If data are not known put it into limitation of the study.

Response: We thank Reviewer 1 for the time that he/she has taken to review our manuscript. We are heartened by his/her positive review. We agree that we should have highlighted the vaccination status of the patients. As many of the patients received vaccinations in other jurisdictions or outside the public hospital system (via their GP) we were not able to reliably access these data. Therefore, in the revised submision, we have added this as a limitation of the paper (as suggested by the reviewer)

Reviewer 2 Report

Comments and Suggestions for Authors

1. Line 44: “rheumatoid arthritis” – please abbreviate (see line 98, 110 etc.) and check abbreviation management in the entire manuscript.

2. Line 46: “traditional disease-modifying anti-rheumatic drugs (DMARDs)” – the correct term is “conventional synthetic disease-modifying anti-rheumatic drugs (csDMARDs)”.

3. At line 109 you write: “they had received b/tsDMARDs for rheumatoid arthritis (RA) or juvenile idiopathic arthritis (JIA)” and at line 113 you write: “Patients were excluded if they were children (age <18 years)”. Why consider eligible patients who you know you will exclude?

4. Line 125: “Australia’s Pharmaceutical Benefits Scheme”. Please note that you aim to publish this article in an international journal, and international readers do not know the legal requirement for RA patients to receive reimbursed b/tsDMARD in Australia. This is relevant since these criteria act as selection filters for more difficult RA cases. The reader must know what RA phenotype you are treating. You only mention joint counts, please briefly mention all the said criteria and their numerical values if any (for example SJC28 > 5).

5. Line 127 and Table 1, 3 “corticosteroids” – the preferred literature term is “glucocorticoids”.

6. Statistical Analysis – please state in this section how you report your continuous variables (means with SDs or medians with IQRs) and based on what.

7. Table 1 - Joint count – what are you reporting by this term, tender joints, swollen joints, both? Please clarify in the Methods and Table.

Author Response

We thank the Reviewer for the time that he/she has taken to review our manuscript. We are delighted to read his/her positive review. Please find below our point-by-point respond to his/her comments.

1. Line 44: “rheumatoid arthritis” – please abbreviate (see line 98, 110 etc.) and check abbreviation management in the entire manuscript.

Response: We thank the reviewer for his/her suggestion. We have abbreviated rheumatoid arthritis to RA throughout the manuscript.

2. Line 46: “traditional disease-modifying anti-rheumatic drugs (DMARDs)” – the correct term is “conventional synthetic disease-modifying anti-rheumatic drugs (csDMARDs)”.

Response: We thank the reviewer for his/her suggestion. We have amended traditional disease-modifying anti-rheumatic drugs (DMARDs) to conventional synthetic disease-modifying anti-rheumatic drugs (csDMARDs) throughout the manuscript.

3. At line 109 you write: “they had received b/tsDMARDs for rheumatoid arthritis (RA) or juvenile idiopathic arthritis (JIA)” and at line 113 you write: “Patients were excluded if they were children (age <18 years)”. Why consider eligible patients who you know you will exclude?

Response: We thank the Reviewer for raising this point. No patients under the age of 18 were included in the study, but there were 10 patients who were diagnosed with juvenile idiopathic arthritis before the age of 16. There were 8 patients who had serological test results available and in 4 this was positive.  All 10 of the patients had a final diagnosis of rheumatoid arthritis - and received b/tsDMARD therapy for a diagnosis of RA - during the study period. We have therefore removed the reference to juvenile idiopathic arthritis in the paper to avoid this confusion.

4. Line 125: “Australia’s Pharmaceutical Benefits Scheme”. Please note that you aim to publish this article in an international journal, and international readers do not know the legal requirement for RA patients to receive reimbursed b/tsDMARD in Australia. This is relevant since these criteria act as selection filters for more difficult RA cases. The reader must know what RA phenotype you are treating. You only mention joint counts, please briefly mention all the said criteria and their numerical values if any (for example SJC28 > 5).

Response: We thank the Reviewer for raising this point and agree that it is an important point to highlight. We have added text to the methods to describe the criteria for prescription of reimbursed b/tsDMARD therapy as suggested by the Reviewer:

“Under Australia’s Pharmaceutical Benefits Scheme patients are only eligible for subsidised b/tsDMARD therapy if they have trialled at least 2 csDMARDs over a 6-month period and have persistent disease activity as determined clinically by their treating rheumatologist. Persistent disease activity is defined by the presence of at least 20 tender or swollen small joints or 4 or more large joints on examination together with raised inflammatory markers (CRP> 15 mg/L and or ESR > 25mm/hour). “

5. Line 127 and Table 1, 3 “corticosteroids” – the preferred literature term is “glucocorticoids”.

Response: We thank the reviewer for his/her suggestion. We have amended corticosteroids to glucocorticoids throughout the manuscript.

6. Statistical Analysis – please state in this section how you report your continuous variables (means with SDs or medians with IQRs) and based on what.

Response: We thank the reviewer for raising this point. We presented three continuous variables (age, Charlon Comorbidity Index and joint count). On visual inspection of a histogram of the distribution of these variables it was clear that 2/3 (Charlson Comorbidity Index and joint count) had a non-parametric distribution. We therefore chose to present the median and interquartile range of all 3 continuous variables (joint count and age). We have added text to the statistical methods to highlight this.

7. Table 1 - Joint count – what are you reporting by this term, tender joints, swollen joints, both? Please clarify in the Methods and Table.

Response: We thank the reviewer for raising this point. Yes, tender and swollen joints were included in the joint count. We have highlighted this in the methods and table as suggested.

Reviewer 3 Report

Comments and Suggestions for Authors

The study is of clear interest to physicians treating rheumatological diseases with b/tsDMARD therapy not only in tropical settings. Understanding treatment patterns across continents will help improve approaches to reducing the incidence of serious, life-threatening infections.

This retrospective cohort study aimed to define the incidence, aetiology, associations and clinical course of serious infections in patients with rheumatoid arthritis receiving b/tsDMARD therapies in this unique region of tropical Australia.

The manuscript contains a well-structured introduction that clearly defines the infectious disease landscape in specific areas of Australia.

This retrospective study has a good design. It is important that patient recruitment occurred prior to the impact of significant COVID-19 transmission in the region.

Demographic data, medical records, laboratory and radiological data before the study were well characterized. Detailed characteristics of the 4 patients who died and the 12 patients admitted to the intensive care unit were presented.

The sub-sections of the “Results” (Indigenous Australian patients, b/tsDMARD Therapy, Screening for Infection Prior to Initiation of b/tsDMARD Therapy, etc) are well highlighted and very interesting.

The list of error comments is below:

1.       It was not possible to determine how many patients in total received DMARDs, of which 310 patients developed infectious complications. It is recommended to indicate this in the "methods" section.

2.       Patients with rheumatoid arthritis (RA) or juvenile idiopathic arthritis (JIA) were included in the study, These are two different conditions, therefore it is recommended to indicate not one RA disease, but both diseases in the results and conclusions of the study. It is possible to propose the term “inflammatory arthritis”, which includes RA and JIA. It is also recommended to correct the abstract.

3.       The results indicate that, 180/310 (58%) patients had an urban residence while 59/310 (19%) lived remotely. it is not clear what happened to the remaining 23% of patients?

Author Response

We thank Reviewer 3 for the time that he/she has taken to review our manuscript and his/her very positive and constructive comments. Please find below our point-by-point reponse to his/her three queries:

  1. It was not possible to determine how many patients in total received DMARDs, of which 310 patients developed infectious complications. It is recommended to indicate this in the "methods" section.

Response: We thank the Reviewer for raising this question. All 310 patients in the cohort received csDMARDS (which we had described as DMARDS in the prior submission) as this is required for the receipt of PBS reimbursed b/tsDMARD therapy (as we now make clearer in the methods). All 310 patients then went on to receive b/tsDMARD therapy during the study period.

As we highlight in section 3.4. Incidence and Timing of Infection, there were 74/310 (24%) individuals who had 147 episodes of serious infection while receiving b/tsDMARD therapy during the study period

  1. Patients with rheumatoid arthritis (RA) or juvenile idiopathic arthritis (JIA) were included in the study, These are two different conditions, therefore it is recommended to indicate not one RA disease, but both diseases in the results and conclusions of the study. It is possible to propose the term “inflammatory arthritis”, which includes RA and JIA. It is also recommended to correct the abstract.

Response: We thank the Reviewer for raising this point. As we noted in our reply to Reviewer 2, no patients under the age of 18 were included in the study, but there were 10 patients who were diagnosed with juvenile idiopathic arthritis before the age of 16. There were 8 patients who had serological test results available and in 4 this was positive.  All 10 of the patients had a final diagnosis (as an adult) of rheumatoid arthritis - and received b/tsDMARD therapy for a diagnosis of RA - during the study period. We have therefore removed the reference to juvenile idiopathic arthritis in the paper to avoid this confusion. We have not used the term "inflammatory arthritis" as this would include other conditions such as psoriatic arthritis and ankylosing spondylitis which were excluded from the study.

  1. The results indicate that, 180/310 (58%) patients had an urban residence while 59/310 (19%) lived remotely. it is not clear what happened to the remaining 23% of patients?

Response: We thank the reviewer for raining this issue. As we highlight in the third paragraph of the methods patients living within 20km of the Cairns central business district were said to have an urban residence, while those living >100km away were said to live remotely. The “missing” 71/310 (23%) lived between 20 and 100 kms from the Cairns central business district. We felt that people living in this region could neither be said to have an urban, nor a remote, address.